behaviour, ecology

*Gasterosteus aculeatus*, *Schistocephalus solidus*, behavioural manipulation, group behaviour, communication

**Author for correspondence:**
Nicolle Demandt
e-mail: n_vand10@uni-muenster.de

# Parasite infection disrupts escape behaviours in fish shoals

Nicolle Demandt[1], Marit Praetz[1], Ralf H. J. M. Kurvers[2], Jens Krause[3,4], Joachim Kurtz[1] and Jörn P. Scharsack[1,5]

[1]Institute for Evolution and Biodiversity, University of Münster, Hüfferstrasse 1, 48149 Münster, Germany
[2]Center for Adaptive Rationality, Max Planck Institute for Human Development, Lentzeallee 94, 14195 Berlin, Germany
[3]Leibniz-Institute of Freshwater Ecology and Inland Fishery, Müggelseedamm 310, Berlin, Germany
[4]Faculty of Life Sciences, Humboldt-Universität zu Berlin, Invalidenstrasse 42, 10115 Berlin, Germany
[5]Thünen Institute for Fisheries Ecology, Herwigstr. 31, 27572 Bremerhaven, Germany

   ND, 0000-0001-9560-1906; RHJMK, 0000-0002-3460-0392; JKr, 0000-0002-1289-2857;
JKu, 0000-0002-7258-459X

Many prey species have evolved collective responses to avoid predation. They rapidly transfer information about potential predators to trigger and coordinate escape waves. Predation avoidance behaviour is often manipulated by trophically transmitted parasites, to facilitate their transmission to the next host. We hypothesized that the presence of infected, behaviourally altered individuals might disturb the spread of escape waves. We used the tapeworm *Schistocephalus solidus*, which increases risk-taking behaviour and decreases social responsiveness of its host, the three-spined stickleback, to test this hypothesis. Three subgroups of sticklebacks were placed next to one another in separate compartments with shelter. The middle subgroup contained either uninfected or infected sticklebacks. We confronted an outer subgroup with an artificial bird strike and studied how the escape response spread through the subgroups. With uninfected sticklebacks in the middle, escape waves spread rapidly through the entire shoal and fish remained in shelter thereafter. With infected sticklebacks in the middle, the escape wave was disrupted and uninfected fish rarely used the shelter. Infected individuals can disrupt the transmission of flight responses, thereby not only increasing their own predation risk but also that of their uninfected shoal members. Our study uncovers a potentially far-reaching fitness consequence of grouping with infected individuals.

## 1. Introduction

To avoid predation, many prey species have evolved collective responses, which allow them to rapidly transfer information about potential predators [1–4]. Information transfer within groups is well known for many species, including insects [5,6], fish [1–3] and birds [7]. Information about a threat can be rapidly transmitted in waves throughout a group of hundreds or even thousands of individuals [8–11]. The spread of information is generally triggered by individual(s) perceiving the predator locally, subsequently generating a cascade when individuals react to the flight responses of other group members [12]. Such information transmission can provide anti-predatory benefits, even for individuals that have not detected the predator themselves [12].

The benefits of social transmission depend critically on the reliability of social cues—as also maladaptive behaviours can be socially transmitted [13]—and on the local coordination between neighbouring individuals [1,7,14]. Hence, the presence of unresponsive individuals can disrupt the transmission of social cues. For example, ant colonies (*Temnothorax nylanderi*) infected with the tapeworm *Anomotaenia brevis* displayed less aggression towards conspecific competitors compared with uninfected colonies [15]. Many parasites, especially those

with complex life cycles, decrease the anti-predatory behaviour of their hosts by reducing hosts' flight responses, to facilitate transmission to the next host [16–20]. Roach (*Rutilus rutilus*) infected with the tapeworm *Ligula intestinalis* occurred more frequently close to banks, which might increase their encounter rate with piscivorous birds, the parasites' final hosts [17]. Rats infected with the protozoan parasite, *Toxoplasma gondii,* showed a decreased risk perception or even an attraction towards cats, the final host of this parasite [18]. *Toxoplasma gondii* can also infect humans, and correlational studies have shown a positive correlation between infections and aggression and risky behaviour in humans [21,22]. Such alteration of the risk-taking behaviour of infected individuals might lead to the non-transmission of important cues to uninfected conspecifics. However, the influence of such misinformation has barely received any attention thus far.

To fill this gap, we used the three-spined stickleback, *Gasterosteus aculeatus*, and its cestode parasite *Schistocephalus solidus*, an important model in ecological and evolutionary parasitology [23,24]. The three-spined stickleback is a small teleost fish occurring in marine and freshwater habitats all across the Northern Hemisphere [25]. Sticklebacks use public information to assess potential predation risks: they observe the behaviour of conspecifics and other fish species [1], and adjust their shoaling behaviour in an adaptive manner to reduce predation risk. In response to a detected predator, sticklebacks can form large shoals of up to hundreds of individuals [26,27].

The tapeworm *S. solidus* is a parasite with a three-host life cycle that frequently infects sticklebacks [25]. *Schistocephalus solidus* reproduces in the gut of its final host, a fish-eating bird. The eggs of the parasite are released into water with the faeces of the bird, where parasite larvae hatch and infect the first intermediate host, a cyclopoid copepod [24]. Infected copepods are then ingested by their obligatory second intermediate host, the three-spined stickleback [23,28]. When *S. solidus* reaches a weight of approximately 50 mg and becomes sexually mature [29], it starts causing distinct changes in the anti-predatory behaviour of infected sticklebacks [26,30]: *S. solidus*-infected sticklebacks increase their risk-taking behaviour by spending more time in the open water and reducing their flight responses to a predator attack from above, thereby facilitating the parasite's transmission to its final host [20,31–34]. Moreover, when satiated, *S. solidus*-infected sticklebacks decrease their social behaviour compared to uninfected conspecifics [35].

In a previous study, we investigated the consequences of *S. solidus*-infected sticklebacks on the risk-taking behaviour of uninfected shoal members. This study showed that when *S. solidus*-infected sticklebacks increased their risk-taking behaviour, uninfected shoal members adjusted their behaviour to the infected sticklebacks when outnumbered [36]. This suggests that shoaling with infected sticklebacks might increase the predation risk of the uninfected conspecifics. However, this study did not investigate how infected sticklebacks influence the transmission of cues within a shoal.

Therefore, the aim of the present study was to test if *S. solidus*-infected sticklebacks disrupt the transmission of cues of flight responses through a shoal. We used a set-up in which three subgroups of four sticklebacks were placed next to one another. In the middle subgroup, we placed either sham-exposed (i.e. control treatment) or *S. solidus*-infected sticklebacks (i.e. infected treatment). We then confronted one of the outer subgroups (the 'stimulus' group) with an artificial bird strike and studied how the flight response spread to the middle ('transmission') and third ('response') subgroup. We hypothesized that the presence of *S. solidus*-infected sticklebacks in the middle of a shoal would dampen the flight response of the uninfected conspecifics the farthest away from the bird attack.

## 2. Material and methods

### (a) Experimental animals
Laboratory-bred F1 offspring of wild-caught three-spined sticklebacks and *S. solidus* parasites, collected, respectively, in April 2017 and September 2017 in the brook Ibbenbürener Aa (Germany, 52°17′33.51″ N 7°36′45.46″ E), were used. F1 stickleback families were obtained by *in vitro* fertilization and housed in family groups in 16 L tanks (VewaTech, Germany) with artificial plants as shelter. Sticklebacks were maintained in recirculating tap water at 17°C with 16 : 8 h day/night cycle with 1 h twilight and fed daily ad libitum with frozen *Chironomid* larvae. Two weeks before the start of the experiment, dry food flakes (Tetra, Germany) were added to the diet to familiarize the sticklebacks with the food stimulus used during the experimental trials.

For parasite reproduction, the hermaphrodite tapeworms were bred *in vitro* [37–39]. We used size-matched pairs of parasites for reproduction as this increases the likelihood of out-crossing, and reduces the likelihood of selfing [39]. Parasite eggs were washed and stored for at least two weeks at 4°C to simulate winter conditions. The eggs were then incubated for three weeks at 20°C in the dark to stimulate larvae development. The hatching of larvae was initiated by illumination and eggs were kept in a 16 : 8 h day/night cycle for two more days. Hatched larvae were individually transferred to individual copepods in wells of 24-well plates with 2 ml tap water. Fourteen days post-exposure, *S. solidus* infection was determined using a microscope.

At nine months of age, sticklebacks ($n = 332$) were gathered from pools consisting of sticklebacks from seven different families and isolated into jars with 400 ml tank water. After 2 days of starvation, the sticklebacks were either offered *S. solidus*-infected copepods ($n = 222$) or uninfected copepods ($n = 110$; sham-exposed sticklebacks for the control treatment). The next day, all glasses were checked for the presence of non-eaten copepods and none were found. All *S. solidus*-exposed fish were placed in three holding tanks of 80 l and all sham-exposed fish were placed in two other same sized holding tanks. After 69 days, the presence of *S. solidus* in the stickleback's body cavity was determined by inspecting bodily swelling of exposed sticklebacks [40]; 98 out of 222 exposed sticklebacks were deemed infected.

After determining the infection status, experimental fish (80 *S. solidus*-infected, 80 sham-exposed and 260 naive sticklebacks gathered from pools containing the same families as the infected and sham-exposed fish) were tagged with visible implant elastomer (VIE) tags (Northwest Marine Technology [41]). VIE tags were used so that infected individuals could be identified back after the experiments and the parasite burden for each individual could be determined. The behavioural experiments started three weeks after tagging.

All animal experimental procedures were executed in accordance with the local veterinary and animal welfare authorities (project number: 84-02.04.2014.A368).

### (b) Experimental set-up
To investigate how the transmission of cues through a shoal is influenced by the presence of infected sticklebacks, we used an experimental tank (47.5 × 7 × 60 cm) consisting of three equally sized compartments (figure 1*a*). The experimental tank was limited to a width of 7 cm to ensure two-dimensional analysis

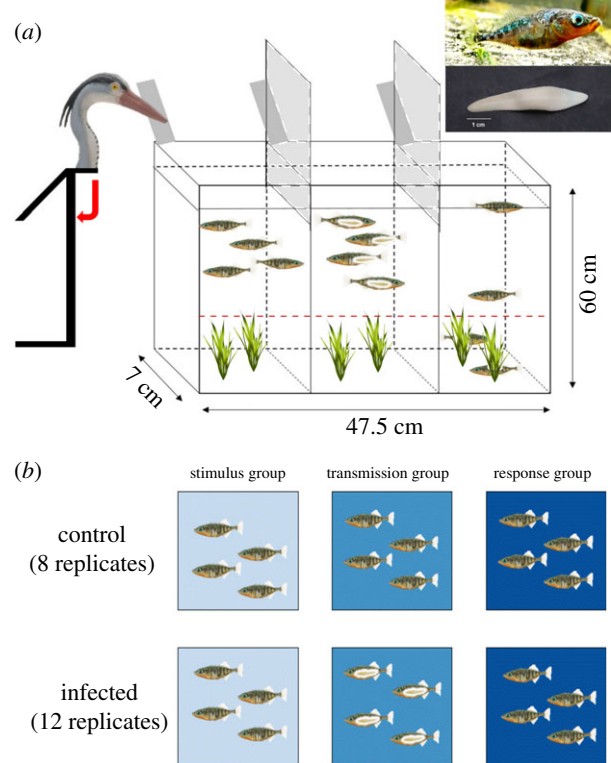

**Figure 1.** (*a*) Schematic overview of the experimental tank (see electronic supplementary material, videos S1 and S2). The tank consisted of three compartments, each containing four fish and two artificial plants providing shelter at the bottom of the tank ('safe' zone). Dimensions are not shown to scale. The red dashed line above the plants indicates the boundary to the 'dangerous' open water zone. Tilted dark-grey rectangles represent food ramps and vertical light-grey squares represent Plexiglas sheets to prevent the detection of the artificial bird by sticklebacks in the other compartments (i.e. the transmission and response subgroup). An artificial bird attached to a stand was used to simulate a bird attack. In the middle subgroup, infected fish are shown. Photographs in the top right show a stickleback and a parasite dissected from its host. (*b*) Experimental design split per treatment, subgroup and infection status. The artificial bird was placed next to the stimulus subgroup. The top row shows the control treatment without infected sticklebacks, while the bottom row shows the infected treatment. (Online version in colour.)

of the fleeing response. In the outer compartments, we placed 'stimulus' and 'response' subgroup, each composed of four naive, uninfected sticklebacks. In the central compartment, we placed a 'transmission' subgroup, consisting of either four sham-exposed sticklebacks ('control') or four infected sticklebacks ('infected'). Both treatments were replicated 12 times resulting in 48 infected, 48 sham-exposed and 192 naive sticklebacks ($n = 288$) (figure 1*b*).

The compartments were divided by two glass walls, allowing visual but not chemical communication (confirmed by pilot experiments). Each compartment was divided into two zones: a 'safe' zone at the bottom (10 cm) with two artificial plants providing shelter and a 'dangerous' upper zone without any shelter (figure 1*a*). Next to the stimulus subgroup, an artificial bird head was installed. The fish were unable to detect the bird beak until it was triggered. For food administration, PVC ramps ($25 \times 2 \times 1$ cm) were installed at the back of each compartment. White Plexiglas covered the back as well as the left and right sides of the tank. To prevent the detection of the artificial bird by sticklebacks in the other compartments (i.e. the transmission and response subgroup), two white Plexiglas sheets ($23 \times 7$ cm) were attached to the top 7 cm of the glass walls dividing the compartments. The set-up was illuminated by a lamp installed above the experimental tank and the light was filtered

through a light filter (Lee no. 251 quarter white diffusion) to reduce light reflection. On the open side of the tank, a Logitech HD Pro C920 webcam was placed to record the trials. The entire set-up was shielded by black cloth so the experimenter could provide the food stimuli and trigger the artificial bird without being noted by the sticklebacks.

## (c) Test procedure and behavioural observations

The evening prior to the experiment, sticklebacks were tagged individually with small plastic discs on the first dorsal spine (to facilitate individual tracking during the experiment) [42] and placed with their subgroup mates in a separate tank. Sticklebacks within a subgroup came from the same holding tank, whereas sticklebacks in different compartments came from different holding tanks. Sticklebacks were not fed on the day of testing in order to increase their feeding motivation.

All sticklebacks within one trial were simultaneously put into the experimental tank and given 15 min to acclimate before the observation started. After 5 min of observation (i.e. before the bird strike), a food stimulus was provided to all compartments. When at least three sticklebacks in the stimulus subgroup and all sticklebacks in the other subgroups had approached the water surface (i.e. within two body lengths), an artificial bird strike was triggered (see electronic supplementary material, videos). Recordings continued for another 10 min after the bird strike to observe the fish during their recovery from the bird strike. Next, all sticklebacks were collected and measured to the nearest millimetre, disc tags were removed and the sticklebacks were placed into a new holding tank. If the requirements for the bird strike were not met within 30 min after acclimatization, the experiment was ended. This occurred four times for the control treatment and never for the infected treatment. After each trial, the experimental tank was cleaned and filled with new water.

The behaviour of each stickleback was analysed using the event logging software BORIS [43]. A mask marking the border between the safe and dangerous zone was placed on the video screen. A fish was considered to be in the dangerous zone when its entire body (including the tailfin) was in the dangerous zone (i.e. crossed the mask). For each stickleback, the time spent in the dangerous zone was recorded 5 min before the bird strike. Once the bird strike was triggered, the zone to which each stickleback escaped was recorded as well as the maximum fleeing depth (cm). After all sticklebacks had stopped their escape behaviour (i.e. when they ceased their rapid downward movement and resumed normal swimming), time spent in the dangerous zone was recorded for another 5 min.

## (d) Parasite burden

To determine whether the parasite burden influenced the behaviour of the infected sticklebacks, all infected sticklebacks were dissected after the last trial. Before dissection, all sticklebacks were weighed to the nearest milligram, anaesthetized by a blow on the head and killed by decapitation. Parasite(s) were dissected out of the body cavity and weighed to the nearest milligram. Parasite burden was calculated as the fraction of the parasite(s) weight over the total weight of the stickleback plus parasite(s).

## (e) Statistical analysis

For statistical analyses, R v. 3.4.4 [44] was used. For all graphs, the function ggplot (package 'ggplot2' [45]) was used.

To analyse the effect of treatment and subgroup (i.e. stimulus, transmission or response) on the likelihood of remaining in the dangerous zone versus escaping to the safe zone after the bird strike, generalized linear mixed models (GLMMs) were used with the function glmer (package 'lme4' [46]) and a binomial distribution with a probit link function. To analyse the effect of

treatment and subgroup on an individual's maximum fleeing depth, we used a linear mixed model (LMM) with the function lmer (package 'lme4' [46]). For both full models, treatment, subgroup and their two-way interaction were fitted as fixed effects. As random intercepts, subgroup ID was nested within trial number to account for the possibility that individuals within one experimental subgroup and trial might behave more similarly to one another than individuals from different subgroups/trials.

To analyse the effect of treatment and subgroup on the time spent in the dangerous zone before and after the bird strike, we used an LMM with the function lme (package nlme [47]) with weights function varComb and varIdent to allow for separate residual variances per treatment and time point. In the full model, treatment, subgroup, bird strike (before/after) and all two- and three-way interactions were included as fixed effects. As random intercepts, fish ID nested in subgroup ID nested within trial number was included to account for repeated testing. Additionally, the length of each stickleback—centred with the function scale (package 'stats' [44])—was included as a covariate in formerly discussed full models.

Finally, using only infected sticklebacks, the effect of parasite burden on the fleeing depth and the time spent in the dangerous zone was analysed using an LMM with the function lmer. Parasite burden was added as a fixed effect to all full models. For time spent in the dangerous zone, we additionally fitted bird strike (before/after) and its interaction with parasite burden. As random intercept, either subgroup ID (i.e. fleeing depth) or fish ID nested in subgroup ID (i.e. time spent in the dangerous zone) were included to account for repeated testing.

In all the above models, the control function (g)lmerControl (calc.derivs = F) was used to avoid the time-consuming derivative calculations.

For all GLMMs, and for LMMs after refitting to maximum likelihood, the significance of the fixed effects was determined using likelihood ratio tests (LRTs) and a stepwise backwards elimination, with the drop1 function (package: 'lme4' [46]), to obtain the minimum adequate models (MaM). After determining the MaM, all LMMs were first refitted to restricted maximum likelihood, the residuals of all models were visually [48] (functions qqPlot, package 'car' [49] and plot, package 'stats' [44]) and statistically inspected for normality (function ad.test, package 'nortest' [50]) and homogeneity (function bartlett.test, package 'stats' [44]). Only for the time in the dangerous zone, deviations in normality and homoscedascity were found. After checking the residuals, *post hoc* tests were performed—using the function emmeans (package 'emmeans' [51])—to determine which specific groups statistically differed.

## 3. Results

### (a) Effects of parasite infection on the flight transmission through a shoal

To investigate if infected sticklebacks in the middle ('transmission') subgroup influenced the escape behaviour of the sticklebacks in the third ('response') subgroup, we recorded for each individual in each subgroup (i.e. stimulus, transmission and response) whether it escaped to the lower safe zone (with shelter) or remained in the upper dangerous zone (without shelter). There was a significant interaction between treatment and subgroup on the likelihood to remain in the dangerous zone (LRT = 7.165, d.f. = 2, $p = 0.028$; figure 2$a$,$b$; electronic supplementary material, S1–S3): *post hoc* tests revealed that fish in the stimulus subgroups of both treatments did not differ in their likelihood to remain in the dangerous zone ($Z = 0.075$, $p = 0.94$). However, fish in both the transmission and response subgroups of the infected treatment

were more likely to remain in the dangerous zone after the bird strike than fish in the corresponding subgroups in the control treatment (transmission: $Z = -2.687$, $p = 0.007$; response: $Z = -2.026$, $p = 0.043$). When comparing the escape behaviour of subgroups within each treatment, we found no difference in escape behaviour between the subgroups in the control treatment (all $p = 1$; figure 2$a$). However, in the infected treatment, fish in both the transmission and response subgroups remained in the dangerous zone more often than the stimulus fish (both $p < 0.01$; figure 2$b$).

Mirroring the above results, there also was a significant interaction between treatment and subgroup on individuals' fleeing depths (LRT = 13.712, d.f. = 2, $p = 0.001$; figure 2$c$,$d$; electronic supplementary material, S1–S3): *post hoc* test showed that fish in the transmission subgroup of the control treatment fled deeper than individuals in the transmission subgroup of the infected treatment ($t = 4.755$, $p < 0.001$). However, fish in the stimulus and response subgroups of both treatments did not differ in their fleeing depths (stimulus: $t = 0.704$, $p = 0.485$ and response: $t = 1.593$, $p = 0.119$). When comparing fleeing depths of subgroups within each treatment, we found no differences between the subgroups in the control treatment (all $p > 0.7$; figure 2$c$). In the infected treatment, fish from the stimulus subgroup fled deeper than fish from the transmission and response subgroups (transmission: $t = 5.920$, $p < 0.001$; response: $t = -2.676$, $p = 0.033$; figure 2$d$). Moreover, the infected fish in the transmission subgroup fled less than the fish in the response subgroup ($t = 3.165$, $p = 0.009$).

### (b) Effects of parasite infection on the time spent in the dangerous zone

To investigate whether the presence of infected sticklebacks in the transmission subgroup influenced the time sticklebacks in the other subgroups spent in the dangerous zone, we studied the time each individual spent in the dangerous zone before and after the bird strike. There was a significant two-way interaction between treatment and bird strike (i.e. before versus after) on the time that individuals spent in the dangerous zone (LRT = 35.707, d.f. = 1, $p < 0.001$; figure 2$e$,$f$; electronic supplementary material, S4–S7), but no effect of subgroup (LRT = 1.598, d.f. = 2, $p = 0.45$; figure 2$e$,$f$; electronic supplementary material, S7). *Post hoc* tests revealed that before the bird strike, fish from both treatments did not differ in their time spent in the dangerous zone ($t = -0.404$, $p = 0.69$; electronic supplementary material, S7). After the bird strike, fish from the infected treatment spent more time in the dangerous zone than fish from the control treatment ($t = -5.006$, $p < 0.001$; figure 2$e$,$f$). Whereas fish in the control treatment reduced their time in the dangerous zone ($t = -7.641$, $p < 0.001$; electronic supplementary material, S7), fish in the infected treatment did not change their time spent in the dangerous zone following the bird strike ($t = -1.061$, $p = 0.29$; electronic supplementary material, S7).

### (c) Effects of parasite burden on behaviour of infected sticklebacks

To investigate if the behaviour of infected sticklebacks depended on their parasite burden, we analysed the relationship between parasite burden and our behavioural measures. A non-significant negative trend was found between parasite burden and fleeing depth after the bird strike (LRT = 3.7442,

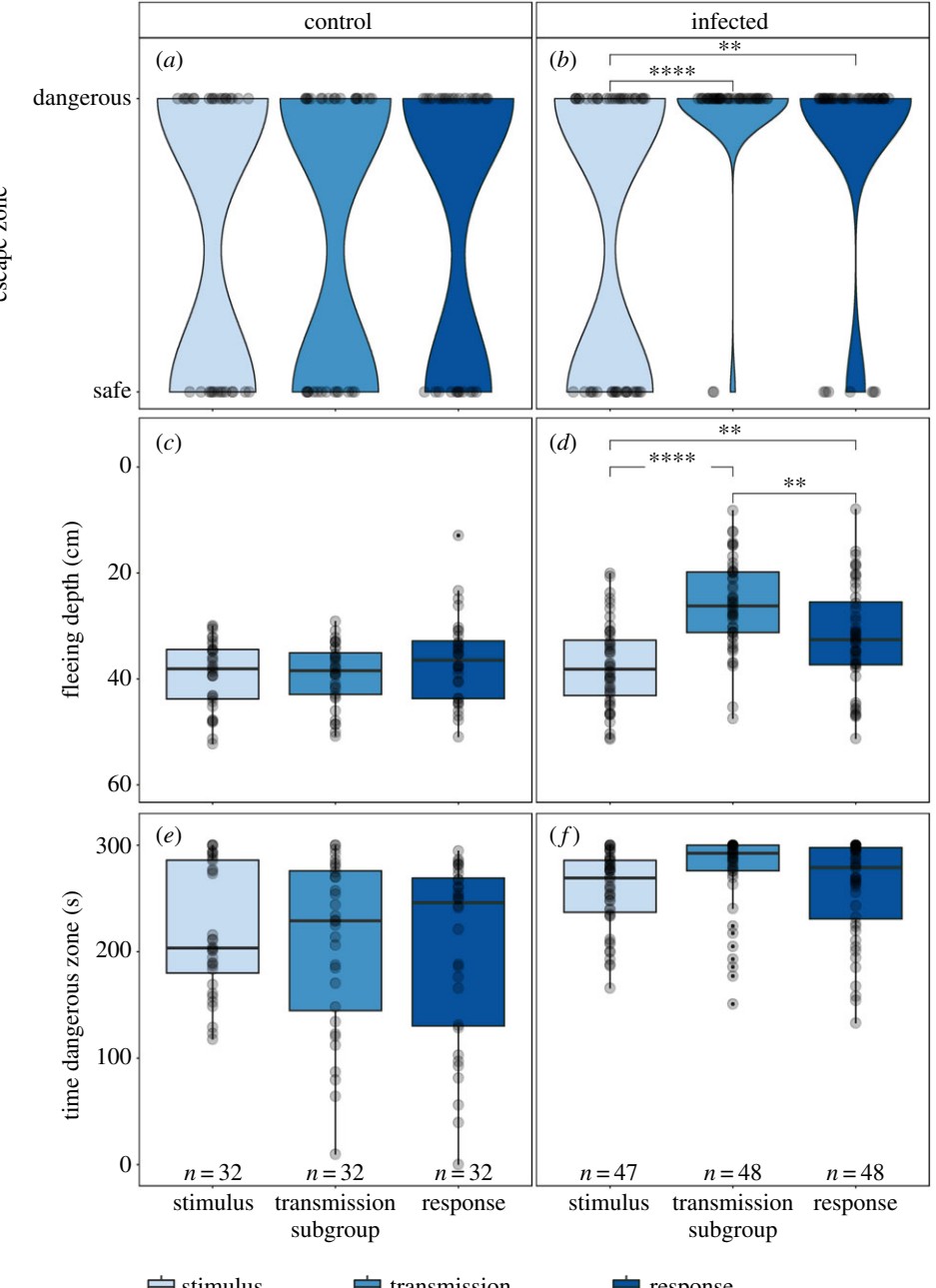

**Figure 2.** Escape responses of sticklebacks to an artificial bird strike (a,c,e) without (control) and (b,d,f) with *S. solidus*-infected conspecifics in the transmission group (infected). (a,b) Sticklebacks escaped to the safe zones at the bottom of the tanks or remained in the upper dangerous zones. The maximum width of the violins is scaled to be constant. (c,d) Maximum fleeing depth of sticklebacks after the bird strike. (e,f) Time spent in the dangerous zone. Results are shown per subgroup: stimulus (light blue), transmission (blue) and response (dark blue) subgroups. The edges of the box plots indicate the first and third quartiles; the solid lines the median, the whiskers the highest and lowest values within 1.5-fold of the inter-quartile range and the transparent dots represent all individual data points. (Online version in colour.)

d.f. = 1, p = 0.05; figure 3a; electronic supplementary material, S8 and S9), suggesting that fish with a higher parasite burden fled less deep than fish with a lower parasite burden. Moreover, independent of bird strike (LRT = 0.086, d.f. = 1, p = 0.77; figure 3b; electronic supplementary material, S8 and S9), fish with a higher parasite burden tended to spent more time in the dangerous zone (LRT = 3.672, d.f. = 1, p = 0.06; figure 3b; electronic supplementary material, S8–9).

## 4. Discussion

The reduction in predation risk is one of the key advantages of group living [7,52]. Successful transmission of information is

an essential prerequisite for adequate group responses to predator attacks [11]. Parasites that manipulate the predator avoidance behaviour of their hosts to facilitate transmission to their next host might interfere with the transmission of public cues in groups. Here, we tested how sticklebacks that were behaviourally altered by infection with the tapeworm *S. solidus* influenced the transmission of cues to uninfected conspecifics. When infected sticklebacks were present in the middle of the group, the uninfected sticklebacks in the response group reduced their fleeing depth and less often escaped to the safe zone at the bottom of the tank in response to an artificial bird strike. After the bird strike, all subgroups in the control treatment reduced their time in the upper dangerous zone of the tank, while subgroups with neighbouring

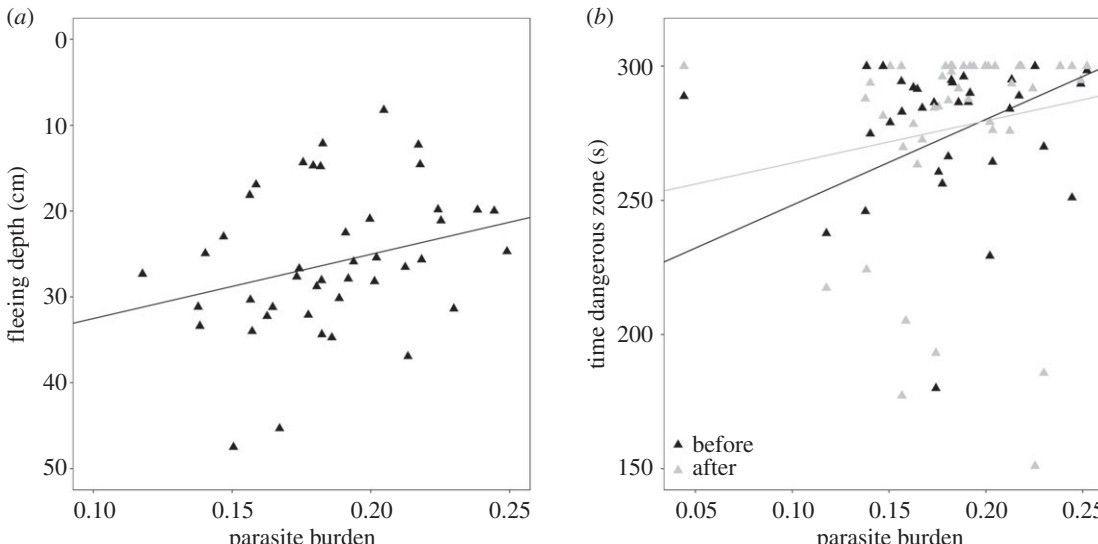

**Figure 3.** The relationship between an individual's parasite burden and its (a) fleeing depth and (b) time spent in 'dangerous' zone. Both panels include only experimentally infected sticklebacks (i.e. transmission subgroups of the infected treatment). In (b), each individual is shown twice: before (black) and after (grey) the bird strike. Lines were extracted from the respective models and plotted manually.

infected sticklebacks did not do this and instead returned to the food stimulus at the water surface much faster. The present study illustrates that parasite-induced behavioural alterations of individual group members can disrupt the transmission of a flight response through a fish shoal, thereby not only increasing their own predation risk but potentially also that of their uninfected shoal members. This could potentially have far-reaching fitness consequences for individuals grouping with infected individuals.

Rapid risk assessment, balancing the benefits of foraging against the risk of predation, and fast decision making when confronted with a potential predator, are observed in many animal species [53], including mammals [54], fish [31,55] and aquatic insects [56]. Therefore, it is likely that the stickle-backs in our study made risk assessments when determining how deep they should flee. In the control treatment with uninfected sticklebacks, the fleeing response spread instantly by (visual) communication and individuals in the subgroup furthest from the artificial bird strike reached the same fleeing depth as those in the stimulus subgroup. However, in the infected treatment, infected sticklebacks in the middle compartment fled less and as a result, individuals in the sub-group furthest from the artificial bird strike fled less than the corresponding subgroup in the control treatment. Accordingly, the uninfected sticklebacks next to infected conspecifics stayed closer to the food at the water surface than response subgroups in the 'control' treatment. By staying at a similar height as the infected sticklebacks, the uninfected ones in the response subgroup maintained their position in the shoal. This might be a strategy to retain some protection by the group from potential further attacks as most predators only capture one prey at a time from a group [57,58]. Further-more, the likelihood of an individual being captured declines with group size, thus fleeing to the depth at which other neigh-bouring individuals are present, might be a safer strategy than separating from the group by fleeing deeper.

Since the energy required for swimming is proportional to the fishes' cube of speed [59], fleeing is energetically costly [60]. Therefore, sticklebacks that stop their downward movement more rapidly will save energy compared with

sticklebacks fleeing deeper. Fleeing might also come with a cost of losing (foraging) opportunities. Consequently, it is likely that the uninfected individuals responded to the changes in speed of their (infected) neighbours, which has been shown to be an important source of information within a shoal [10,61–63], to stop their flight response.

In our experimental set-up, sticklebacks in the 'response' subgroup could not only see the fish in the middle compart-ment ('transmission' subgroup), but also those in the first compartment ('stimulus' subgroup). Future studies could avoid this by using an L-shaped aquarium in which the response subgroup only has visual contact to the 'trans-mission', but not to the 'stimulus' subgroup. However, we decided against this, since we consider that visual contact to the stimulated individuals would better resemble a natural escape situation. Despite our conservative set-up, we still observed a reduced fleeing response of the response sub-group in the infected treatment even though the fish could see the escaping stimulus subgroup. This suggests that the observed mechanism might even be more pronounced in natural habitats where individuals may only observe a few close by neighbours: in natural habitats, visibility might, for example, be lower than in our experimental set-up due to turbidity [64,65], environmental complexity (i.e. algae/rocks) and/or limited light penetration. However, as stickle-backs occur in very different aquatic systems (clear lakes, turbid streams and marine environments) [25], the impli-cations of our results for the wild will depend on the local environment. In clear waters, our results might be directly translated to the natural environment and fish might use vision to respond to cues from other fleeing sticklebacks. In more turbid environments, fish might rely more on other senses that only take in more local information (e.g. the lateral line) [64]. Also in such a scenario, infected individuals might break the line of communication, if they do not react to neighbouring individuals escaping. Other factors that might influence the translation of our laboratory study to the wild are group size, the number of infected individuals and their parasite burden, and the position of the infected sticklebacks within the shoal. Especially, the positioning of the infected

sticklebacks within a shoal might be relevant; therefore, it would be interesting to repeat our experimental set-up with infected sticklebacks in other compartments.

Though behavioural manipulation by parasites is well known [15–20,66], limited attention has been given to how the level of parasite burden influences the extent of the behavioural manipulation. Here, we looked at the link between parasite burden and fleeing depth and time spent in dangerous zone before and after bird strike. Sticklebacks with a higher parasite burden had the tendency to flee less deep and spent more time in the dangerous zone than sticklebacks with a lower parasite burden. This suggests that energetic drain might play a role in the extent of the behavioural manipulation. Parasites extract energy, in the form of nutrition, from their hosts [67], leading to, for example, reduced perivisceral fat reserves [68], and a higher likelihood of dying under poor food conditions [69]. The link between energetic drain and behavioural manipulation has been illustrated for multiple species [67,70,71], and might shift the trade-off between predator avoidance and foraging towards the latter, which increases the host's susceptibility to predators [72].

Another intriguing future research direction is the potential link between the level of parasite burden and the strength of the behavioural response of uninfected conspecifics. In our study, the average parasite burden of the infected transmission subgroups did not explain variation in behaviour in the response subgroups (data not shown). However, our study was not ideal to test this question, and set-ups with only one fish in each compartment using infected individuals differing in parasite burden may prove fruitful.

The present and our former study [36] illustrate the risk of shoaling with *S. solidus*-infected sticklebacks. In natural situations, with real predators, the reduction in anti-predatory responses by uninfected sticklebacks might make them more prone to predation as well. This would have far-reaching consequences for the fitness of individuals grouping together with infected individuals. Our results suggest that these effects might also depend on the parasite burden of the infected individual. Therefore, any negative consequences of shoaling with infected individuals might be lower at the onset of infection. In the wild, uninfected social organisms often co-occur with infected individuals when the parasites do not transfer horizontally [73–75].

If uninfected individuals do not avoid grouping with infected ones, it suggests that the overall benefits of grouping outweigh the costs of infection under certain environmental conditions [74]. For *S. solidus*-infected individuals, it is possible that they are faster at spotting suitable food sources as they have a higher demand for food [23,69], which the uninfected conspecifics can, in turn, exploit as they are better at food competition than the *S. solidus*-infected sticklebacks [31]. This would be especially advantageous in food-limited environments.

With the present study, the disruption of signal transmission due to the presence of infected individuals in groups was studied. These results might be relevant and important for more, if not all, social (prey) species and their vertically transmitted parasites, maybe even humans [21,22].

**Ethics.** All animal experimental procedures were approved by and executed in accordance with the local veterinary and animal welfare authorities (project number: 84-02.04.2014.A368).

**Data accessibility.** The data that support the findings of this study are openly available on the Dryad Digital Repository: https://dx.doi.org/10.5061/dryad.5dv41ns46 [76].

**Authors' contributions.** N.D. and M.P. established the experimental set-up. M.P. performed the behavioural observations of the sticklebacks and did the preliminary analysis of the data. N.D. supervised the practical work and data evaluation, prepared the statistical analysis and wrote the first draft of the manuscript. R.H.J.M.K. provided input on the behavioural analysis and edited each version of the manuscript thoroughly. J.Kr. and J.Ku. conceived of the study together with J.P.S. who supervised the project work. The manuscript was mainly written by N.D., but represents the joint effort of all authors.

**Competing interests.** The authors declare no competing interests.

**Funding.** The project was supported by a DAAD stipend to N.D. (no. 91557716).

**Acknowledgements.** We thank Kathrin Brüggemann, Georg Plenge, Luis Garcia Rodriguez and further members of the workshop of the Faculty of Biology for their technical assistance. We further thank Leonie Grotendorst and Leonie Bley for help with fish husbandry. We are grateful to three anonymous reviewers for their comments, which greatly helped to improve the manuscript.

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
