## [Reviewer comments · Proceedings of the Royal Society B: Biological Sciences]

Review History

RSPB-2020-1158.R0 (Original submission)

Review form: Reviewer 1

Recommendation

Accept with minor revision (please list in comments)

Scientific importance: Is the manuscript an original and important contribution to its field?

Excellent

General interest: Is the paper of sufficient general interest?

Good

Quality of the paper: Is the overall quality of the paper suitable?

Excellent

Is the length of the paper justified?

Yes

Should the paper be seen by a specialist statistical reviewer?

No

Do you have any concerns about statistical analyses in this paper? If so, please specify them explicitly in your report.

No

It is a condition of publication that authors make their supporting data, code and materials available - either as supplementary material or hosted in an external repository. Please rate, if applicable, the supporting data on the following criteria.

Is it accessible?

Yes

Is it clear?

Yes

Is it adequate?

Yes

Do you have any ethical concerns with this paper?

No

Comments to the Author

General comments

This is an interesting experimental study showing that fish are affected by the behavior of infected fish that do not respond to social signals of their shoal members. The findings have important consequences for social animal including humans. The experiments have been well conducted and explained. Maybe a little more information on other systems both in the introduction and discussion would help to widen the angle and to elicit the interest of readers from other fields.

Specific points

Title

The usage of the word animal groups is somewhat misleading, as is escape waves. The fish were separated by glass barriers.

How about Parasite infection disrupts escape behaviors in fish shoals

Abstract

The abstract is nicely writing and conveys the importance of the study and the major findings.

Introduction

The introduction is rather short (only 1 page) it provides information in a condensed way and shortly introduces the experimental design. Maybe a little more information on other group-living organisms and how infection leads to changes in uninfected group members including social birds and mammals and insects. Citations are given, but no details are provided.

Line 70 While the authors use group behavior to describe the behavior of the entire shoal, in the experimental design part, they use group for each of the 4 fish in a part of the tank. Group is both the entire shoal and all subgroups. It might help to call the 4 fish a subgroup to avoid confusion.

Material and Methods

Consider to move parts of the detailed methods to a supplement, allowing for more space in the introduction and discussion.

Line 82 Why did you do in vitro fertilization of sticklebacks? Does that influence in any way the results?

Line 88 Why do size-matched pairs of worms increase outbreeding?

Line 95 Why three to four families?

Line 104 Where did the naive sticklebacks come from? What treatment did the sham-exposed fish receive?

Line 110 Why was the tank only 7 cm in width?

Line 145 How did you recognize whether a fish was in the dangerous zone: Did it have to be there with its entire body, or just parts of it?

Results

The results are generally very well explained and easy to follow.

Discussion

Especially as the authors introduced with the title the effects of group-living organisms, the reader would expect a broader discussion. As it is it focusses more on fish systems and does not go far beyond. There is literature on other social organisms and how infection affects non-infected group members responses such as against predators or enemy, which could be included. Also more literature could be discussed on collective behaviors in fish swarms.

Figure 1

The feeder resembles the shelters, which leads to confusion Please use a drawing of the green plant as shelter, the grey rectangles are weird. Why not include a real photo of the artificial bird and its mechanism to simulate an attack. Please make the upper Plexiglas screens shaded, so it is clear that they are opaque. Could you have an insert showing the fish and the parasite to scale?

Figure 2

Generally well done. The positioning of a), b) and so on is a little weird, Maybe always in the upper part? It would be good to indicate which groups are significantly different from each other? Is it possible to include sample sizes in the figure?

Figure 3

The different shades of grey from the symbols are not very distinct. Consider to use more distinct colours.

Review form: Reviewer 2

Recommendation

Accept with minor revision (please list in comments)

Scientific importance: Is the manuscript an original and important contribution to its field?

Excellent

General interest: Is the paper of sufficient general interest?

Excellent

Quality of the paper: Is the overall quality of the paper suitable?

Excellent

Is the length of the paper justified?

Yes

Should the paper be seen by a specialist statistical reviewer?

No

Do you have any concerns about statistical analyses in this paper? If so, please specify them explicitly in your report.

No

It is a condition of publication that authors make their supporting data, code and materials available - either as supplementary material or hosted in an external repository. Please rate, if applicable, the supporting data on the following criteria.

Is it accessible?

Yes

Is it clear?

Yes

Is it adequate?

Yes

Do you have any ethical concerns with this paper?

No

Comments to the Author

General comments

I really liked this paper, which tackles a novel and interesting question about how well-documented impacts of a parasite that reduces host antipredator behaviour might impact the transmission of information across social groups. It is a study that would have wide interest and appeal, and it is surprising that this has not been tackled before. I found this to be a rigorous and well-designed experimental study that has been logically analysed. The paper is extremely well written, and is clear and succinct, perhaps even too much so, and I think the discussion should be expanded to place the results of this controlled lab study into a clearer ecological context (see below).

As the authors point out in the discussion, given the small size of the aquarium tanks that they used, it seems very likely that the fish in the response group could see and react to not only the fish in the stimulus group but also to the heron itself, as well as fish in the 'transmission' group, and I actually would have expected the tanks to be a little larger to counteract this likelihood. A more sophisticated design might have placed the three tanks to form an 'L' shape, to test more directly the ideas about infected fish in the linking group, but I understand the authors' point about their design perhaps mimicking the natural situation more clearly. The fact that they still have found a significant effect despite this issue convinces me that this is likely an important mechanism that is perhaps even more likely in more naturally lit environments, with reduced visibility than in a laboratory aquarium. This latter fact might be emphasised in the discussion.

Minor comments:

- Line 15-16: "We used the tapeworm *Schistocephalus solidus*, which increases risk-taking behaviour and decrease social responsiveness of its host, the three spined stickleback" needs an ending such as "...to test this hypothesis"
- Line 22: "...fish did hardly use the shelter." Better to say "...fish rarely used the shelter."
- Line 42-43: "Such alteration of the risk-taking behaviour of infected individuals might lead to the transmission of false information to uninfected conspecifics" - I think it would be clearer to say "...might lead to the non-transmission of important information to..."
- Line 104-105 - Why were the fish VIE tagged? Is this relevant to the study? If not, perhaps explain that this was done for husbandry reasons and is not relevant to the study?

- Line 130 – Why were the fish spine tagged – was this for the tracking software to be able to follow the fish? Need to explain the relevance of this.
- Line 165-166 – “In all models the control function (g)lmerControl(calc.derivs=F) was used.” OK but explain why it is important and what the function actually does.
- Discussion (general) – I would appreciate some more discussion of how differences between the environment created in the lab-based study and those of typical natural stickleback habitat (with respect to, e.g., complexity, light levels, turbidity, and so on) affect how the lab results translate into the real-world scenario. What are the new questions that this raises?
- Line 311. I don't think the authors can really leave the tantalising 'even humans' part hanging without providing some relevant reference to an appropriate study that has documented behavioural change in parasitised human hosts, so that the interested reader can investigate that literature further.
- Figure 1a. This is OK but it really would be very easy to include some slightly more realistic-looking artificial plants than just some grey rectangles to represent them...! Also, I feel that the plerocercoids superimposed on the fish might easily get lost when reproduced at smaller size so the authors will need to do something to make it clearer that these are infected fish.

Decision letter (RSPB-2020-1158.R0)

10-Jul-2020

Dear Mrs Demandt:

Your manuscript has now been peer reviewed and the reviews have been assessed by an Associate Editor. The reviewers' comments (not including confidential comments to the Editor) and the comments from the Associate Editor are included at the end of this email for your reference. As you will see, the reviewers and the Editors have raised some concerns with your manuscript and we would like to invite you to revise your manuscript to address them.

Research ethics:

Use of animals and field studies:

Please submit a copy of your revised paper within three weeks. If we do not hear from you within this time your manuscript will be rejected. If you are unable to meet this deadline please let us know as soon as possible, as we may be able to grant a short extension.

Best wishes,
Dr Sasha Dall
mailto: proceedingsb@royalsociety.org

Associate Editor
Board Member: 1
Comments to Author:
Dear Ms Demandt and colleagues,

Thank you for your submission. Two expert reviewers have now assessed your manuscript and their detailed comments are provided below. Both are very positive about your article, they found the manuscript clearly written, the experiment well designed, appropriately analysed and with intriguing results. The consensus is that this will provide a novel and important contribution to our understanding of how parasites influence the transmission of behaviour and spread of information in social groups.

That said, both raise several important criticisms. The Introduction and Discussion are too narrowly focused on this experimental system and would benefit from including, in more detail, examples from a wider range of taxa. Reviewer 2 also suggests that the Discussion could go into more detail on generalising from this laboratory experiment to the natural context, and what limitations there might be in such comparisons.

Both reviewers provide several detailed suggestions about how to improve clarity of methodology and presentation of results. For example, more explanation could be provided, earlier on, for why such small tanks are used for the study and Reviewer 2 would like to see more emphasis on the issue that, given this design, it is possible for fish in the 'response' group to be able to observe those in the 'stimulus' group in the Discussion. I agree and think that some of the justification at line. 275 onwards could be made earlier, even when explaining the study set-up.

From my own reading of the article, I agree with all the reviewers' points and have some minor additional suggestions, and one more major point:

Minor

- Do take care when using language about information, particularly in context of 'reliability of information': see Fawcett & Frankenhuis 2015 *Frontiers in Zoology*, Glossary on page 2, which explains that 'information' is uncertainty reduction (always informative), whereas 'cues' can be reliable or unreliable.
- Figure 1 - I do not see a red dashed line above the grey plants? Consider explaining in the legend that the Figure a) shows infected fish in the central zone (otherwise it is not clear what the blobs are in these fish). It would be good to refer to the video in the ESM here and/or in the main text. This will then help readers understand why the grey shapes are used for plants: both reviewers prefer to see more 'plant-like' shapes drawn here, but if you want to reflect realistic set-up and leave these grey shapes as they are, then it is worth pointing the reader to the video.
- Figure 2 - It may be worth overlaying post hoc comparison results here (e.g., using asterisks) to emphasise which groups were significantly different and which were not.
- Figure 3 - Consider clarifying in legend that these measures are for fish only in the 'response' tanks, whereas bird strike is experienced by fish in neighbouring 'stimulus' tanks
- 1.114 The independent replicate of n=12 trials seems quite a small sample size, I appreciate that there are n=4 fish in each trial/treatment combination, but were any power analyses conducted or other constraints which determined this sample size, and not a larger one?
- 1.139 Please give rationale for recording observations up to 10 minutes post-strike.

Major:

- I was surprised that analysis was only made of consequences of parasite burden for the infected individual's behaviour (discussed at 1.285), and not also the behaviour of uninfected conspecifics (given that the latter is the main focus of this study). Indeed, this is implied at 1.299-300, but not measured in the data. It may be that, given that there are 4 fish in each trial each with varying degrees of parasite burden, one cannot correlate parasite burden of the individual infected fish with response of the neighbouring fish, but if some signal of this was found even when considering mean levels per trial, this would certainly be interesting. Perhaps for this study it suffices to say that the data do not allow addressing this consequence of parasite burden, but more could be made of how this might be measured in future work.

Reviewer(s)' Comments to Author:

Referee: 1

Comments to the Author(s)

General comments

This is an interesting experimental study showing that fish are affected by the behavior of infected fish that do not respond to social signals of their shoal members. The findings have important consequences for social animal including humans. The experiments have been well conducted and explained. Maybe a little more information on other systems both in the introduction and discussion would help to widen the angle and to elicit the interest of readers from other fields.

Specific points

Title

The usage of the word animal groups is somewhat misleading, as is escape waves. The fish were separated by glass barriers.

How about Parasite infection disrupts escape behaviors in fish shoals

Abstract

The abstract is nicely writing and conveys the importance of the study and the major findings.

Introduction

The introduction is rather short (only 1 page) it provides information in a condensed way and shortly introduces the experimental design. Maybe a little more information on other group-living organisms and how infection leads to changes in uninfected group members including social birds and mammals and insects. Citations are given, but no details are provided.

Line 70 While the authors use group behavior to describe the behavior of the entire shoal, in the experimental design part, they use group for each of the 4 fish in a part of the tank. Group is both the entire shoal and all subgroups. It might help to call the 4 fish a subgroup to avoid confusion.

Material and Methods

Consider to move parts of the detailed methods to a supplement, allowing for more space in the introduction and discussion.

Line 82 Why did you do in vitro fertilization of sticklebacks? Does that influence in any way the results?

Line 88 Why do size-matched pairs of worms increase outbreeding?

Line 95 Why three to four families?

Line 104 Where did the naive sticklebacks come from? What treatment did the sham-exposed fish receive?

Line 110 Why was the tank only 7 cm in width?

Line 145 How did you recognize whether a fish was in the dangerous zone: Did it have to be there with its entire body, or just parts of it?

Results

The results are generally very well explained and easy to follow.

Discussion

Especially as the authors introduced with the title the effects of group-living organisms, the reader would expect a broader discussion. As it is it focusses more on fish systems and does not go far beyond. There is literature on other social organisms and how infection affects non-infected group members responses such as against predators or enemy, which could be included. Also more literature could be discussed on collective behaviors in fish swarms.

Figure 1

The feeder resembles the shelters, which leads to confusion Please use a drawing of the green plant as shelter, the grey rectangles are weird. Why not include a real photo of the artificial bird and its mechanism to simulate an attack. Please make the upper Plexiglas screens shaded, so it is clear that they are opaque. Could you have an insert showing the fish and the parasite to scale?

Figure 2

Generally well done. The positioning of a), b) and so on is a little weird, Maybe always in the upper part? It would be good to indicate which groups are significantly different from each other? Is it possible to include sample sizes in the figure?

Figure 3

The different shades of grey from the symbols are not very distinct. Consider to use more distinct colours.

Referee: 2

Comments to the Author(s)

General comments

I really liked this paper, which tackles a novel and interesting question about how well-documented impacts of a parasite that reduces host antipredator behaviour might impact the transmission of information across social groups. It is a study that would have wide interest and appeal, and it is surprising that this has not been tackled before. I found this to be a rigorous and well-designed experimental study that has been logically analysed. The paper is extremely well written, and is clear and succinct, perhaps even too much so, and I think the discussion should be expanded to place the results of this controlled lab study into a clearer ecological context (see below).

As the authors point out in the discussion, given the small size of the aquarium tanks that they used, it seems very likely that the fish in the response group could see and react to not only the fish in the stimulus group but also to the heron itself, as well as fish in the 'transmission' group, and I actually would have expected the tanks to be a little larger to counteract this likelihood. A more sophisticated design might have placed the three tanks to form an 'L' shape, to test more directly the ideas about infected fish in the linking group, but I understand the authors' point about their design perhaps mimicking the natural situation more clearly. The fact that they still have found a significant effect despite this issue convinces me that this is likely an important mechanism that is perhaps even more likely in more naturally lit environments, with reduced visibility than in a laboratory aquarium. This latter fact might be emphasised in the discussion.

Minor comments:

- Line 15-16: "We used the tapeworm *Schistocephalus solidus*, which increases risk-taking behaviour and decrease social responsiveness of its host, the three spined stickleback" needs an ending such as "...to test this hypothesis"
- Line 22: '...fish did hardly use the shelter.'" Better to say "...fish rarely used the shelter.'

- Line 42-43: “Such alteration of the risk-taking behaviour of infected individuals might lead to the transmission of false information to uninfected conspecifics” – I think it would be clearer to say ‘...might lead to the non-transmission of important information to...’
- Line 104-105 – Why were the fish VIE tagged? Is this relevant to the study? If not, perhaps explain that this was done for husbandry reasons and is not relevant to the study?
- Line 130 – Why were the fish spine tagged – was this for the tracking software to be able to follow the fish? Need to explain the relevance of this.
- Line 165-166 – “In all models the control function (g)lmerControl(calc.derivs=F) was used.” OK but explain why it is important and what the function actually does.
- Discussion (general) – I would appreciate some more discussion of how differences between the environment created in the lab-based study and those of typical natural stickleback habitat (with respect to, e.g., complexity, light levels, turbidity, and so on) affect how the lab results translate into the real-world scenario. What are the new questions that this raises?
- Line 311. I don’t think the authors can really leave the tantalising ‘even humans’ part hanging without providing some relevant reference to an appropriate study that has documented behavioural change in parasitised human hosts, so that the interested reader can investigate that literature further.
- Figure 1a. This is OK but it really would be very easy to include some slightly more realistic-looking artificial plants than just some grey rectangles to represent them...! Also, I feel that the plerocercoids superimposed on the fish might easily get lost when reproduced at smaller size so the authors will need to do something to make it clearer that these are infected fish.

Author's Response to Decision Letter for (RSPB-2020-1158.R0)

See Appendix A.

RSPB-2020-1158.R1 (Revision)

Review form: Reviewer 2

Recommendation

Accept as is

Scientific importance: Is the manuscript an original and important contribution to its field?

Excellent

General interest: Is the paper of sufficient general interest?

Excellent

Quality of the paper: Is the overall quality of the paper suitable?

Excellent

Is the length of the paper justified?

Yes

Should the paper be seen by a specialist statistical reviewer?

No

Do you have any concerns about statistical analyses in this paper? If so, please specify them explicitly in your report.

No

It is a condition of publication that authors make their supporting data, code and materials available - either as supplementary material or hosted in an external repository. Please rate, if applicable, the supporting data on the following criteria.

Is it accessible?

Yes

Is it clear?

Yes

Is it adequate?

Yes

Do you have any ethical concerns with this paper?

No

Comments to the Author

I retain the same view as my original review. I think this is a really interesting study and one that is suitable for publishing in PRSB. The authors have done a thorough job of addressing the issues that I raised and this has improved the paper further.

Review form: Reviewer 3

Recommendation

Major revision is needed (please make suggestions in comments)

Scientific importance: Is the manuscript an original and important contribution to its field?

Good

General interest: Is the paper of sufficient general interest?

Good

Quality of the paper: Is the overall quality of the paper suitable?

Acceptable

Is the length of the paper justified?

Yes

Should the paper be seen by a specialist statistical reviewer?

Yes

Do you have any concerns about statistical analyses in this paper? If so, please specify them explicitly in your report.

Yes

It is a condition of publication that authors make their supporting data, code and materials available - either as supplementary material or hosted in an external repository. Please rate, if applicable, the supporting data on the following criteria.

Is it accessible?

No

Is it clear?

N/A

Is it adequate?

N/A

Do you have any ethical concerns with this paper?

No

Comments to the Author

In the text, the authors use GroupID as a random effect but do not use IndividualID. In the appendix IndividualID is a random effect but not in all analyses. There is substantial variation in individual behaviour as they acknowledge. Accounting for the variation due to individuals would almost certainly reduce the effect size of the treatment (an effect known as shrinkage). I highly commend Ch. 11-17 of Gelman and Hill Data Analysis using regression and multilevel/hierarchical models. The methods are not presented clearly enough and the results of the log-linear analyses are completely messed up because of some misunderstanding. The expectation for a Chi-square is its degrees of freedom, but in both the main text and the appendix this doesn't appear to be the case. For example --- line 211 Chi-square 7.165, df= 9, p = 0.028. --- The correct p-value is 0.38 .

Why are there no random effects for Table S4?

Decision letter (RSPB-2020-1158.R1)

18-Sep-2020

Dear Mrs Demandt:

Your manuscript has now been peer reviewed and the reviews have been assessed by an Associate Editor. The reviewers' comments (not including confidential comments to the Editor) and the comments from the Associate Editor are included at the end of this email for your reference. As you will see, the reviewers and the Editors have raised some concerns with your manuscript and we would like to invite you to revise your manuscript to address them.

Research ethics:

Use of animals and field studies:

It is a condition of publication that you make available the data and research materials supporting the results in the article (<https://royalsociety.org/journals/authors/author-guidelines/#data>). Datasets should be deposited in an appropriate publicly available repository and details of the associated accession number, link or DOI to the datasets must be included in the Data Accessibility section of the article (<https://royalsociety.org/journals/ethics-policies/data-sharing-mining/>). Reference(s) to datasets should also be included in the reference list of the article with DOIs (where available).

Please submit a copy of your revised paper within three weeks. If we do not hear from you within this time your manuscript will be rejected. If you are unable to meet this deadline please let us know as soon as possible, as we may be able to grant a short extension.

Best wishes,
Dr Sasha Dall
Editor, Proceedings B
mailto:proceedingsb@royalsociety.org

Associate Editor
Board Member: 1
Comments to Author:

The revised manuscript has been appraised by one of the original reviewers and an independent statistical referee. The reviewer is satisfied with this revised version. The statistical referee, however, raises several important concerns and I agree that these need to be clarified for the paper to be suitable for Proceedings B.

In particular, the authors should be clear on the following issues when reporting their statistics:

- How do they account for repeated measures on the same individual in their analyses comparing the same fish before and after a bird strike (e.g. 1.188-190)? Here one would expect Fish Identity to be included as a random term, but this does not seem to be the case.
- Please be more clear on the phrasing of the statistics results, for example the Chi-sq statistic is from a Likelihood ratio test, so it may make more sense to use the term LRT instead. Similarly, it should be clear what the t-value refers to (from the post-hoc comparison) as mixing Chi-sq and t statistics from the same analysis is confusing (e.g. 1.210-213). Please also be consistent in terminology for random effects ('subgroup ID and trial no.' in main text, versus 'Group ID' and 'Experimental ID' in appendix).

I had a quick look at the data on Dryad myself and suggest the authors provide a meta-data file to explain what the column headings mean, it is difficult to replicate the analysis otherwise. Moreover, they should follow the journal guidelines in terms of providing the code used to generate the statistics and figures: <https://royalsociety.org/journals/authors/author-guidelines/#data>

Minor points:

1.240 - word missing 'in the dangerous ZONE'

1.291-334 - This paragraph is far too long. It should be possible to split, maybe at 1.311.

Fig 2a and 2b: The units of this panel are not clear ('escape zone'), and are the bars indicating numbers of individuals in each area? What does the shape of these bars correspond to?

Reviewer(s)' Comments to Author:

Referee: 3

Comments to the Author(s)

In the text, the authors use GroupID as a random effect but do not use IndividualID. In the appendix IndividualID is a random effect but not in all analyses. There is substantial variation in individual behaviour as they acknowledge. Accounting for the variation due to individuals would almost certainly reduce the effect size of the treatment (an effect known as shrinkage). I highly commend Ch. 11-17 of Gelman and Hill Data Analysis using regression and multilevel/hierarchical models. The methods are not presented clearly enough and the results of the log-linear analyses are completely messed up because of some misunderstanding. The expectation for a Chi-square is its degrees of freedom, but in both the main text and the appendix this doesn't appear to be the case. For example --- line 211 Chi-square 7.165, $df=9$, $p=0.028$. --- The correct p-value is 0.38 .

Why are there no random effects for Table S4?

Referee: 2

Comments to the Author(s)

I retain the same view as my original review. I think this is a really interesting study and one that is suitable for publishing in PRSB. The authors have done a thorough job of addressing the issues that I raised and this has improved the paper further.

Author's Response to Decision Letter for (RSPB-2020-1158.R1)

See Appendix B.

Decision letter (RSPB-2020-1158.R2)

09-Oct-2020

Dear Mrs Demandt

I am pleased to inform you that your manuscript RSPB-2020-1158.R2 entitled "Parasite infection disrupts escape behaviour in fish shoals" has been accepted for publication in Proceedings B.

The AE has recommended publication, but also suggests some minor revisions to your manuscript. Therefore, I invite you to respond to the comments and revise your manuscript. Because the schedule for publication is very tight, it is a condition of publication that you submit the revised version of your manuscript within 7 days. If you do not think you will be able to meet this date please let us know.

[http://datadryad.org/submit?journalID=RSPB&manu=\(Document not available\)](http://datadryad.org/submit?journalID=RSPB&manu=(Document not available)) which will take you to your unique entry in the Dryad repository. If you have already submitted your data to dryad you can make any necessary revisions to your dataset by following the above link.

Please see <https://royalsociety.org/journals/ethics-policies/data-sharing-mining/> for more details.

Sincerely,
Dr Sasha Dall
Editor, Proceedings B
<mailto:proceedingsb@royalsociety.org>

Associate Editor:

Comments to Author:

The authors have done a thorough job of revising the manuscript in response to statistical concerns raised during the most recent round of reviews, and I would agree to include fish ID as a random term for the repeated-measures analysis.

I only have some very minor suggestions:

L150 - Change 'Videos' to lower case 'videos'

L181 - Change 'similar' to 'similarly'

L191 - 'was analysed, we used' -> sentence would read better if this is changed to 'was analysed using'

L199 - 'using likelihood ratio tests' should be followed by '(LRTs), following a [Chi-square -- as symbol] distribution,'

Author's Response to Decision Letter for (RSPB-2020-1158.R2)

See Appendix C.

Decision letter (RSPB-2020-1158.R3)

12-Oct-2020

Dear Mrs Demandt

I am pleased to inform you that your manuscript entitled "Parasite infection disrupts escape behaviour in fish shoals" has been accepted for publication in Proceedings B.

Open Access

Paper charges

Sincerely,

Appendix A

Dear Editors,

Thank you for allowing us to resubmit our manuscript, entitled “Parasite infection disrupts escape behaviours in fish shoals”. We thank the referees and associate editor for their valuable comments, which greatly helped us to improve our manuscript. In short, we (i) substantially broadened the scope of our manuscript by adding (both in the introduction and discussion) more examples and information on other social animals and their parasites; (ii) adjusted the title to better reflect our main findings and (iii) improved all our main figures, closely following the reviewers’ suggestions. Please find below (*in italics*) our detailed responses to all queries, the manuscript with “tracked changes” and the figure legends marked with “tracked changes”.

We hope our revised version is acceptable for publication in Proceedings of the Royal Society B.

With kind regards, on behalf of all authors,

Nicolle Demandt.

Editor

Thank you for your submission. Two expert reviewers have now assessed your manuscript and their detailed comments are provided below. Both are very positive about your article, they found the manuscript clearly written, the experiment well designed, appropriately analysed and with intriguing results. The consensus is that this will provide a novel and important contribution to our understanding of how parasites influence the transmission of behaviour and spread of information in social groups.

That said, both raise several important criticisms. The Introduction and Discussion are too narrowly focused on this experimental system and would benefit from including, in more detail, examples from a wider range of taxa. Reviewer 2 also suggests that the Discussion could go into more detail on generalising from this laboratory experiment to the natural context, and what limitations there might be in such comparisons.

Reply: We agree with the observation that our original manuscript was a bit narrow in its scope. In response, we have added more examples of parasites in other (social) host species to the introduction and discussion. Moreover, we added a section to the discussion on the generalisation from the lab experiment to the natural habitats (see line 303-316).

Both reviewers provide several detailed suggestions about how to improve clarity of methodology and presentation of results. For example, more explanation could be provided, earlier on, for why such small tanks are used for the study and Reviewer 2 would like to see more emphasis on the issue that, given this design, it is possible for fish in the 'response' group to be able to observe those in the 'stimulus' group in the Discussion. I agree and think that some of the justification at line. 275 onwards could be made earlier, even when explaining the study set-up.

Reply: We have clarified the presentation of the methods and results. In the methods section we now, for example, explain why we used such small tanks (namely to be able to track downward fleeing in 2D, L. 124-125). To clarify the results we now, for example, adjusted all the main figures providing a better visualization of the experimental set-up. We further added

a part to the discussion to emphasize the fact that the fish in the response subgroup are able to observe the fish in the Stimulus subgroup (L. 294 -304).

From my own reading of the article, I agree with all the reviewers' points and have some minor additional suggestions, and one more major point:

Minor issues

- Do take care when using language about information, particularly in context of 'reliability of information': see Fawcett & Frankenhuis 2015 *Frontiers in Zoology*, Glossary on page 2, which explains that 'information' is uncertainty reduction (always informative), whereas 'cues' can be reliable or unreliable.

Reply: We adjusted the usage of the term information accordingly, by replacing it by cues when appropriate.

- Figure 1 - I do not see a red dashed line above the grey plants? Consider explaining in the legend that the Figure a) shows infected fish in the central zone (otherwise it is not clear what the blobs are in these fish). It would be good to refer to the video in the ESM here and/or in the main text. This will then help readers understand why the grey shapes are used for plants: both reviewers prefer to see more 'plant-like' shapes drawn here, but if you want to reflect realistic set-up and leave these grey shapes as they are, then it is worth pointing the reader to the video.

Reply: We added a red dashed line above the plants and replaced the grey rectangles by artificial plants, as both reviewers requested. To clarify that the fish in the middle group are the infected fish, we added a sentence to the Figure legend ("In the middle subgroup infected fish are shown") and included two small additional pictures showing a sticklebacks and the actual parasite. Lastly, we refer to the supplementary videos in the figure legend and the methods section (L. 153).

- Figure 2 - It may be worth overlaying post hoc comparison results here (e.g., using asterisks) to emphasise which groups were significantly different and which were not.

Reply: We thank the editor for this suggestion. We added asterisks to indicate significant differences among the subgroups within each treatment (i.e. control and infected). We decided to not include asterisks between groups of different treatments as this made the figure too cluttered.

- Figure 3 - Consider clarifying in legend that these measures are for fish only in the 'response' tanks, whereas bird strike is experienced by fish in neighbouring 'stimulus' tanks

Reply: Please note that this figure only included infected individuals, hence individuals from the transmission groups in the infected treatment. To clarify this, we added the following sentence to the legend: "Both panels only include experimentally infected sticklebacks (i.e. transmission subgroups of the infected treatment)"

-I.114 The independent replicate of n=12 trials seems quite a small sample size, I appreciate that there are n=4 fish in each trial/treatment combination, but were any power analyses conducted or other constraints which determined this sample size, and not a larger one?

*Reply: Before starting the experiments, we conducted a power analysis in R to determine the minimum numbers of replicates per treatment. For this power analysis we used a previous experiment on information transmission (Verheijen FJ (1956). Transmission of a flight reaction amongst a school of fish and the underlying sensory mechanisms. *Experientia* 12:202–204. <https://doi.org/10.1007/BF02170796>) to determine the delta. This analysis indicated that 9 replicates should suffice. However, as a precaution for some replicates not meeting the requirements of a successful experiment, we increased the number of replicates to 12.*

Even if we would not have done this power analysis, we would still have been limited by the number of infected fish, as it takes months of preparation to experimentally ‘create’ infected fish in the lab. For this process, parasites, copepods and fish have to be bred. Copepods have to be exposed to the parasites, and infected copepods have to be fed to the sticklebacks, after which a minimum of eight weeks is needed before one can visually determine the infection status of the. Considering the fact that the infection rate of copepods is at best about 25% and the infection rate of fish exposed to one singly infected copepod is max. 40%, over 400 copepods had to be exposed months before the actual experiments to ensure around 50 infected sticklebacks. Moreover, all copepods have to be manually exposed to parasites and two weeks later checked for parasites, which is also a very time-consuming task.

-l.139 Please give rationale for recording observations up to 10 minutes post-strike.

Reply: We wanted to give fish the opportunity to fully recover from the bird strike before ending the experiments. Based on pilot experiments, we observed that after 10 minutes most of the fish had returned to the dangerous zone at least once, even without infected fish being present. We added an explanatory part to the sentence stating: “to observe the fish during their recovery from the bird strike” (L. 152).

Major:

- I was surprised that analysis was only made of consequences of parasite burden for the infected individual's behaviour (discussed at l.285), and not also the behaviour of uninfected conspecifics (given that the latter is the main focus of this study). Indeed, this is implied at l.299-300, but not measured in the data. It may be that, given that there are 4 fish in each trial each with varying degrees of parasite burden, one cannot correlate parasite burden of the individual infected fish with response of the neighbouring fish, but if some signal of this was found even when considering mean levels per trial, this would certainly be interesting. Perhaps for this study it suffices to say that the data do not allow addressing this consequence of parasite burden, but more could be made of how this might be measured in future work.

Reply: Thank you for this interesting suggestion. We ran an additional analysis correlating the mean parasite burden within the transmission group and the escape behaviours of uninfected conspecifics in the response group. However, we did not find any significant correlations. However, the power of this analysis was very low as they only contained 12 data points, and averaging the parasite burden among the four infected fish decreases the between group variation in mean parasite load. However, these are certainly interesting suggestions for follow-up research, which could be done by using smaller groups of fish. We added a paragraph on this subject to our discussion (L. 330 -335).

Referee 1:

This is an interesting experimental study showing that fish are affected by the behavior of infected fish that do not respond to social signals of their shoal members. The findings have important consequences for social animal including humans. The experiments have been well

conducted and explained. Maybe a little more information on other systems both in the introduction and discussion would help to widen the angle and to elicit the interest of readers from other fields.

Specific points

Title

The usage of the word animal groups is somewhat misleading, as is escape waves. The fish were separated by glass barriers. How about Parasite infection disrupts escape behaviors in fish shoals

Reply: We agree with this assessment and changed the title accordingly.

Abstract

The abstract is nicely writing and conveys the importance of the study and the major findings.

Introduction

The introduction is rather short (only 1 page) it provides information in a condensed way and shortly introduces the experimental design. Maybe a little more information on other group-living organisms and how infection leads to changes in uninfected group members including social birds and mammals and insects. Citations are given, but no details are provided.

Reply: We thank the reviewer for this comment. We now provide several more examples from different taxa of (i) how parasites change the behaviour of their infected host, and (ii) how infected individuals might change the behaviour of uninfected group members (L. 41 – 50)

Line 70 While the authors use group behavior to describe the behavior of the entire shoal, in the experimental design part, they use group for each of the 4 fish in a part of the tank. Group is both the entire shoal and all subgroups. It might help to call the 4 fish a subgroup to avoid confusion.

Reply: We agree that our wording might lead to some confusion and, therefore, changed the wording accordingly throughout the manuscript.

Material and Methods

Consider to move parts of the detailed methods to a supplement, allowing for more space in the introduction and discussion.

Reply: As the editor requested more experimental details, the reviewer was unclear which section to move, and the reviewer asks for more methodological clarifications in his/her following comments, we decided to not remove any parts of the methods. In this way, the reader does not need to switch between the appendix and the main text to understand the experiment. We did, however, expand the intro and discussion as suggested by the reviewer.

Line 82 Why did you do in vitro fertilization of sticklebacks? Does that influence in any way the results?

Reply: We did not want to use wild-caught fish as potential differences in life-histories might then influence the results. In vitro fertilization was used for practical reasons as mating sticklebacks in the lab is quite laborious and often unsuccessful. For in vitro fertilizations, we always randomly assorted the mating pairs to maximize genetic variation in the lab. Whether

there would a difference in results between *in vitro* and *in vivo* fertilization is difficult to answer.

Line 88 Why do size-matched pairs of worms increase outbreeding?

Reply: The parasites we used are hermaphrodites that can reproduce by outbreeding and selfing. Lüscher & Milinski (2003) showed that worms differing in weight were more likely to reproduce by selfing than size-matched worms. As selfing comes at a cost of fitness and a possible reduction in infection rates, we aimed to minimize the chance of selfing by using size-matched pairs of worms. We added this information to the methods (L. 96-98).

Line 95 Why three to four families?

Reply: This comment made us realise that our wording was a bit misleading. The experimental fish were reared in groups of three or four families. For the eventual experiments these fish were then pooled, leading to a total of seven families used. We changed the wording to: "from pools consisting of sticklebacks from seven different families" (L. 105).

Line 104 Where did the naive sticklebacks come from? What treatment did the sham-exposed fish receive?

Reply: the naïve sticklebacks came from pools containing the same families as the infected and sham-exposed fish came from. We added this information to the methods (L. 114-115). The sham-exposed fish received the same treatment as the infected sticklebacks, with the difference that the sham-exposed sticklebacks got an uninfected copepod to eat (see additional clarifications in L.107).

Line 110 Why was the tank only 7 cm in width?

Reply: this was done so that we could track the escape response in 2D. Using wider tanks would have allowed fish to flee both vertically as horizontally. As this was not our main question, we decided for a narrow tank of 7 cm width (often a 2D horizontal tank is limited to a water depth of 5-7cm). We added this clarification to the description of the experimental set-up (L. 124-125).

Line 145 How did you recognize whether a fish was in the dangerous zone: Did it have to be there with its entire body, or just parts of it?

Reply: A mask was placed on top of the laptop screen indicating the dangerous zone. When a fish fully crossed this line (including its tail fin), we recorded it as being in the dangerous zone. We clarified this in the methods section (L. 161-163).

Results

The results are generally very well explained and easy to follow.

Discussion

Especially as the authors introduced with the title the effects of group-living organisms, the reader would expect a broader discussion. As it is it focusses more on fish systems and does not go far beyond. There is literature on other social organisms and how infection affects non-infected group members responses such as against predators or enemy, which could be included. Also more literature could be discussed on collective behaviors in fish swarms.

Reply: we thank the reviewer for this comment. As stated above, we changed the title to better reflect the main findings of our work. To broaden the discussion, we more examples on host-parasite interactions in social organisms, including a section on the translation of our findings to the field, and a section discussing potential density dependent responses of parasite burden.

Figure 1

The feeder resembles the shelters, which leads to confusion Please use a drawing of the green plant as shelter, the grey rectangles are weird. Why not include a real photo of the artificial bird and its mechanism to simulate an attack. Please make the upper Plexiglas screens shaded, so it is clear that they are opaque. Could you have an insert showing the fish and the parasite to scale?

Reply: thank you for your suggestions, we changed the rectangles into artificial plants and shaded the upper Plexiglas screens to make it clearer that they are opaque. We further added a picture of a sticklebacks and its parasite. The image of the artificial bird is actually produced from a photograph taken from the "plastic heron model" that we used for our experiments. We transferred the head to a metal construction, to illustrate the movement of the beak.

Figure 2

Generally well done. The positioning of a), b) and so on is a little weird, Maybe always in the upper part? It would be good to indicate which groups are significantly different from each other? Is it possible to include sample sizes in the figure?

Reply: We changed the positioning of the a), b) etc. to a more consistent location at the top of each panel. We included the sample sizes in panel e) and f) as they were the same for all behaviours. Lastly, we added asterisks to indicate significant differences among subgroups within treatments.

Figure 3

The different shades of grey from the symbols are not very distinct. Consider to use more distinct colours.

Reply: We increased the colour contrast and the size of the symbols, to make it easier to distinguish the symbols.

Referee 2:

I really liked this paper, which tackles a novel and interesting question about how well-documented impacts of a parasite that reduces host antipredator behaviour might impact the transmission of information across social groups. It is a study that would have wide interest and appeal, and it is surprising that this has not been tackled before. I found this to be a rigorous and well-designed experimental study that has been logically analysed. The paper is extremely well written, and is clear and succinct, perhaps even too much so, and I think the discussion should be expanded to place the results of this controlled lab study into a clearer ecological context (see below).

Reply: Thank you for these positive comments, and this interesting suggestion. We added a section to the discussion in which we discuss our results in a broader ecological context (L. 303 - 316).

As the authors point out in the discussion, given the small size of the aquarium tanks that they

used, it seems very likely that the fish in the response group could see and react to not only the fish in the stimulus group but also to the heron itself, as well as fish in the ‘transmission’ group, and I actually would have expected the tanks to be a little larger to counteract this likelihood. A more sophisticated design might have placed the three tanks to form an ‘L’ shape, to test more directly the ideas about infected fish in the linking group, but I understand the authors’ point about their design perhaps mimicking the natural situation more clearly. The fact that they still have found a significant effect despite this issue convinces me that this is likely an important mechanism that is perhaps even more likely in more naturally lit environments, with reduced visibility than in a laboratory aquarium. This latter fact might be emphasised in the discussion.

Reply: Yes, we agree that an L-shaped tank would also have been very interesting, but we decided to simulate a more natural situation in which fish could see all shoal members. We added the L-shaped suggestion to the discussion, and discuss potential implications of reduced visibility in natural habitats compared to settings (L. 293 -316).

Minor comments:

- Line 15-16: “We used the tapeworm *Schistocephalus solidus*, which increases risk-taking behaviour and decrease social responsiveness of its host, the three spined stickleback” needs an ending such as “...to test this hypothesis”

Reply: We added the suggested ending.

- Line 22: ‘...fish did hardly use the shelter.’ Better to say “...fish rarely used the shelter.’

Reply: Done

- Line 42-43: “Such alteration of the risk-taking behaviour of infected individuals might lead to the transmission of false information to uninfected conspecifics” – I think it would be clearer to say ‘...might lead to the non-transmission of important information to...”

Reply: We agree, and changed the sentence accordingly. Please note that we also replaced the word information by cue at several instances -- also in this sentence -- whenever cue was more appropriate (see our reply to first minor comment of the Editor).

- Line 104-105 – Why were the fish VIE tagged? Is this relevant to the study? If not, perhaps explain that this was done for husbandry reasons and is not relevant to the study?

Reply: Fish were VIE tagged so that the infected individuals could be identified after the experiments ended and the parasite load of each individual could be determined. We added a brief explanation to the main text (L. 116-117). The VIE tags were, however, too small for tracking the fish during the experiments. Therefore, during the experiment, we used disc tags during the experiments for tracking. Disc tags, on the other hand, are not suitable for long term monitoring as they relatively quickly came off.

- Line 130 – Why were the fish spine tagged – was this for the tracking software to be able to follow the fish? Need to explain the relevance of this.

Reply: As stated above, fish were spine tagged to identify each individual during the experiments and to facilitate tracking. We added a brief explanation to the text (L. 145).

- Line 165-166 – “In all models the control function (g)lmerControl(calc.derivs=F) was used.” OK but explain why it is important and what the function actually does.

Reply: We added a short description explaining what this function does (L. 182-183).

- Discussion (general) – I would appreciate some more discussion of how differences between the environment created in the lab-based study and those of typical natural stickleback habitat (with respect to, e.g., complexity, light levels, turbidity, and so on) affect how the lab results translate into the real-world scenario. What are the new questions that this raises?

Reply: This is a very interesting suggestion. We added a large section to the discussion in which we discuss the implications of our results for different natural habitats (L. 303-316).

- Line 311. I don't think the authors can really leave the tantalising 'even humans' part hanging without providing some relevant reference to an appropriate study that has documented behavioural change in parasitised human hosts, so that the interested reader can investigate that literature further.

Reply: Fair point, we added two references showing a correlation between risk-taking behaviours and parasite manipulation in humans and added more information on the studies to the introduction (L. 46-48).

- Figure 1a. This is OK but it really would be very easy to include some slightly more realistic-looking artificial plants than just some grey rectangles to represent them...! Also, I feel that the plerocercoids superimposed on the fish might easily get lost when reproduced at smaller size so the authors will need to do something to make it clearer that these are infected fish.

Reply: we replaced the grey rectangles by artificial plants. We further added a clarifying text about the infected middle group and added a picture of a stickleback and its parasite to the figure.

Appendix B

Dear Editor,

Thank you for allowing us to resubmit our manuscript, we thank the statistical referee and associate editor for their valuable comments, which helped us improve our manuscript. In short, we (i) re-analysed the time spent in dangerous zone data with a model including fish ID to control for repeated measurements; (ii) adjusted the LRT values in the results sections and the visual representation of Fig. 2a, b; (iii) improved the statistical analysis section in the methods and (iv) added a README file to Dryad and included all R codes in the manuscript.

Please find below (*in italics*) our detailed responses to all queries and the manuscript, figure legends and ESM marked with “tracked changes”.

We hope our revised version is acceptable for publication in Proceedings of the Royal Society B.

With kind regards, on behalf of all authors,

Nicolle Demandt.

Associate Editor

The revised manuscript has been appraised by one of the original reviewers and an independent statistical referee. The reviewer is satisfied with this revised version. The statistical referee, however, raises several important concerns and I agree that these need to be clarified for the paper to be suitable for Proceedings B.

In particular, the authors should be clear on the following issues when reporting their statistics:
- How do they account for repeated measures on the same individual in their analyses comparing the same fish before and after a bird strike (e.g. l.188-190)? Here one would expect Fish Identity to be included as a random term, but this does not seem to be the case.

Reply: We agree, and apologize for the mistake, in the first draft, we compared the model with and without FishID to one another and compared the AIC values to determine which model had the best fit. In this case it was the model without FishID. However, it is common sense and as stated by both the statistical referee and the editor our models should account for repeated measures. Therefore, we re-analysed the variables that included a before and after bird strike observation (i.e. time in dangerous zone) and adjusted everything accordingly. The addition of FishID did not change the results. However, the model to analyse the effect of treatment, time point and group on the time spent in the dangerous zone experienced issues with respect to homoscedascity when Fish ID was included that including FishID. To control for this, we added a weights function to allow for differences in variances, however it did not solve the issue completely as there was still some heterogeneity among the treatments.

- Please be more clear on the phrasing of the statistics results, for example the Chi-sq statistic is from a Likelihood ratio test, so it may make more sense to use the term LRT instead. Similarly, it should be clear what the t-value refers to (from the post-hoc comparison) as mixing Chi-sq and t statistics from the same analysis is confusing (e.g. l210-213). Please also be consistent in terminology for random effects ('subgroup ID and trial no.' in main text, versus 'Group ID' and 'Experimental ID' in appendix).

Reply: We changed the Chi-square to LRT as suggested by the editor. We further, added a reference to the post hoc tests before starting to discuss the results from the post hoc tests to clarify where the

t and z values came from. Due to the comments of the statistical referee, we also carefully checked the appendix and statistics again and realized that the post hoc tests of the log-linear analyses should have referred to a z value. We corrected this accordingly. Lastly, we changed the terminology of the random effects in the appendix, so that they would match the terminology in the main text.

I had a quick look at the data on Dryad myself and suggest the authors provide a meta-data file to explain what the column headings mean, it is difficult to replicate the analysis otherwise. Moreover, they should follow the journal guidelines in terms of providing the code used to generate the statistics and figures: <https://royalsociety.org/journals/authors/author-guidelines/#data>

Our reply: We added a readMe file to the data in Dryad and added the R code for the statistics and figures to the manuscript.

Minor points:

l240 - word missing 'in the dangerous ZONE'

Reply: We adjusted it accordingly.

l291-334 - This paragraph is far too long. It should be possible to split, maybe at l.311.

Reply: We acknowledge this and started a new paragraph as suggested at L.311 and started a new paragraph at L. 322.

Fig 2a and 2b: The units of this panel are not clear ('escape zone'), and are the bars indicating numbers of individuals in each area? What does the shape of these bars correspond to?

Our reply: Yes, the violins are indicating the number of individuals in each area. To clarify this we added the individual data points into the graphs. We further adjusted the legend to clarify the shape of the bars. It now states: "The maximum width of the violins is scaled to be constant."

Reviewer(s)' Comments to Author:

Referee: 3

Comments to the Author(s)

In the text, the authors use GroupID as a random effect but do not use IndividualID. In the appendix IndividualID is a random effect but not in all analyses. There is substantial variation in individual behaviour as they acknowledge. Accounting for the variation due to individuals would almost certainly reduce the effect size of the treatment (an effect known as shrinkage). I highly commend Ch. 11-17 of Gelman and Hill Data Analysis using regression and multilevel/hierarchical models.

Reply: We acknowledge the input of the referee and adjusted all models in which we had multiple observation per individual (i.e. the models containing bird strike) accordingly. Unfortunately, adding Fish ID to the models lead to deviations in homoscedascity, so we choose for an LME model with weights functions varCom and varIdent to allow for separate residual variances per treatment and time point. This did not solve the homoscedascity issue completely, but it improved the model greatly.

The methods are not presented clearly enough and the results of the log-linear analyses are completely messed up because of some misunderstanding. The expectation for a Chi-square is its degrees of freedom, but in both the main text and the appendix this doesn't appear to be the case. For example --- line 211 Chi-square 7.165, df= 9, p = 0.028. --- The correct p-value is 0.38 .

Reply: Thank you for your comment. We corrected the Df accordingly and further refer to LRT instead of Chi-square. We further realized that we used the estimates instead of the t and z values, and corrected the log-linear analyses. Where we accidentally wrote t-value instead of z-value for the post hoc tests.

Why are there no random effects for Table S4?

Reply: Table S4 is the table to test the significance of the models. We did not test for the significance of the random effects.

Referee: 2

Comments to the Author(s)

I retain the same view as my original review. I think this is a really interesting study and one that is suitable for publishing in PRSB. The authors have done a thorough job of addressing the issues that I raised and this has improved the paper further.

Appendix C

Dear Editor,

Thank you for accepting our manuscript for publication in Proceedings of the Royal Society B. We adjusted the minor revisions and added the manuscript marked with “tracked changes” below.

With kind regards, on behalf of all authors,
Nicolle Demandt

Associate Editor

Comments to Author:

The authors have done a thorough job of revising the manuscript in response to statistical concerns raised during the most recent round of reviews, and I would agree to include fish ID as a random term for the repeated-measures analysis.

I only have some very minor suggestions:

I.150 - Change 'Videos' to lower case 'videos'

I.181 - Change 'similar' to 'similarly'

I.191 - 'was analysed, we used' -> sentence would read better if this is changed to 'was analysed using'

I.199 - 'using likelihood ratio tests' should be followed by '(LRTs), following a [Chi-square -- as symbol] distribution,'

Reply: We implemented all minor suggestions in the manuscript.